# Isolation and Characterization of *Trichoderma* spp. for Antagonistic Activity against Avocado (*Persea americana* Mill) Fruit Pathogens

María Estela López-López [1], Carmen Lizette Del-Toro-Sánchez [2,*], Melesio Gutiérrez-Lomelí [1,*], Salvador Ochoa-Ascencio [3], José Antonio Aguilar-López [4], Miguel Angel Robles-García [1], Maribel Plascencia-Jatomea [2], Ariadna Thalia Bernal-Mercado [2], Oliviert Martínez-Cruz [2], María Guadalupe Ávila-Novoa [1], Jean Pierre González-Gómez [5] and Pedro Javier Guerrero-Medina [1]

1 Microbial and Food Biotechnology Research Center, Cienega University Center, University of Guadalajara, University Avenue 1115, Ocotlan 47820, Mexico
2 Department of Research and Postgraduate Studies in Food, University of Sonora, Rosales and Niños Heroes Avenue S/N, Hermosillo 83000, Mexico
3 Faculty of Agrobiology "President Juárez", Michoacan University of San Nicolás de Hidalgo, Uruapan Campus, Paseo Lázaro Cárdenas 2290, Emiliano Zapata, Melchor Ocampo, Uruapan 60170, Mexico
4 Coordination of Food Genomics, University of Ciénega del Estado de Michoacán de Ocampo, Universidad Avenue 3000, Lomas de la Universidad, Sahuayo 59103, Mexico
5 National Laboratory for Food Safety Research, Food and Development Research Center, Carretera a Eldorado Km 5.5, Culiacan 80110, Mexico
* Correspondence: carmen.deltoro@unison.mx (C.L.D.-T.-S.); melesio.gutierrez@academicos.udg.mx (M.G.-L.)

**Abstract:** In this research, we aimed to isolate and identify native strains of *Trichoderma* spp. with potential activity against avocado pathogens (*Neofusicoccum parvum*, *Colletotrichum gloeosporioides*, *Diaporthe* sp., and *Phomopsis perseae*). Strains of *Trichoderma* spp. were isolated from roots and soil obtained from avocado orchards from different regions of Mexico. Twenty-five *Trichoderma* spp. strains were isolated, of which six (TSMICH7, TSMICH8, TRMICH9, TSMICH10, TSMICH15, and TRJAL25) showed greater antagonistic capacity in vitro (>80%) against avocado pathogens. After 96 h, the antagonistic strain undergoes a thickening of hyphae, while the phytopathogen tends to thin, except for *Diaporthe* sp., which tends to thicken. The characterization of these strains was carried out through morphological observations and the amplification and sequencing of rDNA fragments (ITS regions), as well as the translation elongation factor 1-alpha (*Tef 1-α*), achieving the identification of *Trichoderma harzianum*. However, in the in vivo evaluation (applying directly to the avocado fruit), the TSMICH7 strain maintained considerably high effectiveness (>90%) against the four phytopathogens tested, mainly with *P. perseae*, *N. parvum*, and *Diaporthe* sp., increasing the activity of glucanases and chitinases. Therefore, *T. harzianum* could be used as a biological control agent to inhibit post-harvest pathogens in avocados, thus avoiding significant losses of this fruit of international importance.

**Keywords:** *Trichoderma harzianum*; avocado phytopathogens; chitinases; glucanases

## 1. Introduction

Biological control is essential in plant disease management practices associated with phytopathogens [1]. Traditionally, phytopathogen management is facilitated using agrochemicals, which are mainly toxic to the environment and human wellness. The excessive use of agrochemicals has caused deleterious environmental effects due to the contamination of soils and water sources, human health, and the biodiversity of microorganisms and beneficial insects for the crop [2]. Therefore, it is necessary to reduce the environmental impact derived from food production, leading to modern agriculture and maximizing productivity by using biological alternatives that ensure profitability for the farmer [2].

Infections associated with phytopathogens are responsible for 20–40% of total production losses caused by plant diseases [3]. Phytopathogen management includes resistant varieties, use of pesticides, and biological control [4]. The central objective of using biocontrol microorganisms is to limit the application of agrochemicals, reducing the presence of phytopathogens in an environmentally friendly way [4]. Among the biocontrol products registered in the Environmental Protection Agency (EPA), the most used microorganisms in the listed products correspond to fungi of the genus *Trichoderma* spp., particularly *T. viride*, *T. harzianum,* and *T. lignorum* [4]. Its use in agriculture is due to mechanisms of action such as competition, mycoparasitism, antibiosis, and the production of volatile compounds that reduce the infection of plant pathogens [5,6]. For example, some studies have shown the effectiveness of *Trichoderma* spp. against *Botrytis cinerea* in strawberry [7]; *Alternaria alternata*, *Colletotrichum gloeosporioides*, and *Penicillium digitatum* in orange [8]; *Fusarium proliferatum* in apple [9]; *Fusarium oxysporum* in banana [10]; *Colletotrichum* sp. and *Fusarium* sp. in mangrove [11]; and *Fusarium incarnatum* in muskmelon [12], among others. Through the analysis of *Trichoderma's* activities, it has been determined that each species produces different types of lytic enzymes that differ in their activity and efficiency in degrading the cell wall of pathogenic fungi, such as glucanases and chitinases [13].

The cultivation of avocado (*Persea americana* Miller) is of great nutritional and economic importance for Mexico [14]. Mexico is the world's largest producer of avocado, with a volume close to 2.2 million tons. The largest producer is the state of Michoacán, with 79.1% of the total production, followed by Jalisco (9.5%) [15]. Post-harvest diseases that damage the fruit play a crucial role in avocado marketing since they impact the fruit's quality. The principal diseases of avocado fruit caused by various microorganisms are anthracnose or body rot (BR), peduncular rot or stem-end rot (SER), and black spots. In general, the primary pathogens are *Colletotrichum gloeosporioides* and some fungi of the family Botryosphaeriaceae, *Neofusicoccum parvum*, *Diaporthe* sp., and *Phomopsis perseae* [16,17].

Due to the regulations and prohibitions on using fungicides in avocado-importing countries, biocontrol has surged as a very well-known alternative in Mexico, with low risk and effective control of plant diseases, becoming a part of cultural practices in preharvest and routine post-harvest stages [18]. There are studies on the activity of different species of *Trichoderma*; however, when these species are applied to crops other than those from which they were isolated, their effect is not the same, reducing their activity. For this reason, it is crucial to evaluate native strains of *Trichoderma* from the same avocado crops to control pathogenic fungi in the environmental conditions of production [19]. Therefore, the objective of this research work was to study the biological control of *N. parvum*, *C. gloeosporioides*, *Diaporthe* sp., and *P. perseae* in post-harvest avocado fruits (*P. americana* Mill) using *Trichoderma* spp. isolated directly from avocado orchards.

## 2. Materials and Methods

### 2.1. Plant Material and Phytopathogens

Avocado fruits harvested in 2018 were used as plant material in their state of physiological maturity (21.5% dry matter, according to the limits established by the Mexican Standard NMX-FF-016-SCFI-2006) obtained from different avocado orchards (Table 1). Phytopathogens *N. parvum*, *C. gloeosporioides*, *Diaporthe* sp., and *P. perseae* were obtained from the strain collection of the Phytopathology Laboratory of the Faculty of Agrobiology (Michoacan University of San Nicolás de Hidalgo, Mexico).

**Table 1.** Characteristics of the sampling places for the isolation of *Trichoderma* spp.

| Sample Code | Sampling Location (Orchard) | City | State | Coordinates | m.a.s.l. [1] | Season | Particular Characteristics |
|---|---|---|---|---|---|---|---|
| TSMICH1 TSMICH2 TSMICH3 TSMICH4 TSMICH5 TSMICH6 TSMICH7 | Rancho Las Ayacatas | Uruapan (Jucutacato) | Michoacan | Latitude (19°22′39″ N) Longitude (103°55′13″ W) | 2186 | Fall–winter | Avocado variety Hass. Crop age: 30 to 35 years. Use of chemicals. |
| TSMICH8 TSMICH9 TSMICH10 | Rancho El Durazno | Uruapan (Jucutacato) | Michoacan | Latitude (19°4′52″ N) Longitude (102°20′38″ W) | 2329 | Fall–winter | Avocado variety Hass. Crop age: 30 to 35 years. Use of chemicals. |
| TSMICH11 TSMICH12 TSMICH13 TSMICH14 TSMICH15 TSMICH16 TSMICH17 TSMICH18 TSMICH19 TSMICH20 TSMICH21 TSMICH22 | Rancho Villa de las Flores | Uruapan (Jucutacato) | Michoacan | Latitude (19°4′52″ N) Longitude (102°20′38″ W) | 2329 | Fall–winter | Avocado variety Hass. Crop age: 30 to 35 years. Use of chemicals. |
| TRJAL23 TRJAL24 TRJAL25 | Rancho Las Palmas | San Francisco de Asis, Location of Atotonilco, El Alto | Jalisco | Latitude (20°65′17″ N) Longitude (103°25′46″ W) | 1963 | Spring–summer | Avocado varieties Hass and Fuerte. Crop age: 25 to 30 years. 100% organic. Interspersed with fruit trees such as citrus, papaya, walnut, vine, peach, among others. |

[1] m.a.s.l., meters above sea level.

### 2.2. Isolation of Trichoderma spp. from Avocado Orchards

This assay was carried out according to the methodology as previously described [20]. One hundred and fifty grams of roots and soil were sampled (taking from the first 15 cm of depth) from the orchards mentioned in Table 1. Fragments of 1 cm$^2$ were taken from the roots and disinfected with 2% sodium hypochlorite for 3 min. The samples were washed with sterile distilled water, placed in PDA Petri dishes supplemented with chloramphenicol (0.025 mg/mL), and incubated at 28 °C for 2 to 4 days. Next, from the growths obtained, monosporic cultures were made from hyphal tips, obtaining from this process an axenic strain established in a new Petri dish containing PDA. For isolates from the soil, 10 g were weighed and suspended in 90 mL of sterile distilled water. From this mixture, a 1:1000 *v/v* dilution was made. Twenty microliters of the dilution were taken and placed on PDA with 0.025 mg/mL chloramphenicol using the spread plate technique. Monoconidial cultures were performed as indicated previously [21] to ensure the authenticity and purity of the strains obtained.

### 2.3. Characterization and Identification of Trichoderma spp. Strains Isolated

The isolated fungi strains were characterized using morphological and microscopical characteristics [22]. In addition, molecular techniques confirmed strain identification.

#### 2.3.1. Morphological and Microscopical Characterization

Macroscopic characteristics of the colony (radial growth, mycelial color, presence of diffusible pigments, concentric rings) and microscopic characteristics (shape and size of conidiophores, phialides, conidia, and chlamydospores) were traditionally examined with an optical microscope (Model CX311RTSF, Olympus, Tokyo, Japan), using 40× magnification with 40 measurements.

2.3.2. Molecular Identification

Molecular identification was based on PCR amplification of fragments of the DNA region (ITS regions) and the translation elongation factor 1-alpha (*Tef 1-α* gene). Genomic DNA extraction was performed using the protocol described previously [23]. For PCR amplification, the oligonucleotides reported by Chakraborty et al. [24] were used for the rDNA region [ITS1 (5′-TCCGTAGGTGAACCTGCGG–3′) and ITS4 (5′-TCCTCCGCTTATTGATAT GC–3′)], and those reported by Komon-Zelazowska et al. [25] were used for the *Tef 1-α* gene [tef 728f (5′-CATCGAGAAGTTCGAGAAGG–3′) and *Tef 1-α* 1R gene (5′-GCCATCCTTGGG AGATACCAGC–3′]. In each PCR reaction, 20 ng of DNA, 1× buffer, 3 mmol/L of MgCl, 0.2 mmol/L of dNTPs, 0.6 μM of each primer, and 0.2 U of Taq polymerase were used for a final volume of 20 μL. The conditions for amplification for both fragments included initial denaturation at 95 °C for 5 min, followed by 35 cycles at 94 °C for 1 min, 55 °C for 2 min, 72 °C for 1 min, and a final extension at 72 °C for 10 min. The amplified DNA fragments were separated and observed in 2% agarose gel electrophoresis and purified using the PureLink^TM Quick Gel Extraction and PCR Purification kit (Invitrogen, Waltham, MA, USA). Subsequently, the purified fragments were sequenced in both directions (LANGEBIO—CINVESTAV Laboratory, Irapuato, Guanajuato, Mexico), and the sequences were compared for identity with the nucleotide sequences of reported fungi from the National Center Biotechnology Information (NCBI) database using the BLASTN tool (available at: http://www.ncbi.nlm.nih.gov/BLAST (accessed on 4 June 2022)). The alignment of the sequences obtained in this study and the most represented *Trichoderma* species in GenBank was carried out using MUSCLE v3.5 (Berkeley, CA, USA). Subsequently, a maximum likelihood phylogenetic tree was built using PHYML v2.2.3 (Auckland, New Zealand) and visualized in iTOL v6 (Heidelberg, Germany). The ITS and *Tef 1-α* sequences of *Escovopsis clavata* strain LESF853 were used as roots.

*2.4. In Vitro Evaluation of the Antagonistic Capacity of Trichoderma spp.*

Confrontation tests were conducted to evaluate the inhibition exerted by the strains of *Trichoderma* spp. on the phytopathogens (*N. parvum*, *C. gloeosporioides*, *Diaporthe* sp., and *P. perseae*), following the protocol previously described with some modifications [26]. Monoconidial strains were activated at 28 °C for 96 h in a PDA medium. Discs of mycelium (5 mm in diameter) of *Trichoderma* spp. were placed in PDA at one end of the Petri dish, and the phytopathogenic fungi were placed on the opposite side and incubated at 28 °C for 7 days. The tests were performed in triplicate, considering the growth of each strain individually as controls. Readings were taken every 12 h to determine the time of the first contact between the antagonist hyphae and the phytopathogen. The growth of both colonies (cm) was measured, and the percentage of inhibition of radial mycelial growth (IRM) was calculated based on the following formula according to Ibarra-Medina et al. [26]:

$$\%\text{IRM} = \frac{\text{Growth of the antagonist strain (cm)}}{6.5} * 100$$

where 6.5 is the distance in cm between the strain *Trichoderma* spp. and the phytopathogen. Additionally, we conducted a morphometric study to understand the microscopical changes in the structure of both organisms when confronting them. This study consisted of determining the average diameter of hyphae before and after confrontation (*Trichoderma*/Phytopathogen) using an optical microscope (Model CX311RTSF, Olympus, Tokyo, Japan) integrated into an Infinity 1 camera (Lumenera Corp., Ottawa, ON, Canada). This assay was taken with a 40× objective and analyzed using Image Pro-Plus ver. 6.3 software (Media Cybernetics, Inc., Bethesda, MD, USA). At least 60–100 measurements of the diameter of the hyphae were measured from each treatment.

*2.5. In Vivo Evaluation of the Antagonistic Capacity of Trichoderma spp.*

*Trichoderma* conidia were obtained to carry out inhibition studies on avocado fruit. *Trichoderma* spp. was cultivated on PDA at 28 °C for 7 days. Subsequently, 10 mL of

sterile Tween 80 at a concentration of 0.1% was added to the Petri dish and streaked with a Drigalsky spatula. The conidia suspension was then filtered through a polyester fiber to remove mycelial debris. The conidia were counted in a Neubauer chamber and kept in Tween 80 at a concentration of 0.01% at 4 °C until use. The phytopathogens were only cultivated in PDA at 25 °C for 7 days.

Subsequently, the protocol described by Janisiewicz [27] was used with some modifications to execute the in vivo assay. *Trichoderma* spp. ($1 \times 10^6$ conidia/mL) was inoculated (25 μL) in the peduncular area of avocado fruit with physiological maturity, previously disinfested (immersion in 2% sodium hypochlorite solution for 3 min and rinsed with sterile distilled water). The inoculation of the pathogens was performed by placing a 5 mm PDA disc. Subsequently, they were incubated for 7 days at 18 ± 2 °C at 95% relative humidity. At the end of the incubation period, the effect of the inoculum in the peduncular area was evaluated by comparing longitudinal wounds in the fruits and the percentage of damage based on scales of rotting of the peduncle and fruit body [28] (Figure S1). The experiment was divided into four groups in triplicate as follows. Treatment 1: Control (*Trichoderma* spp. conidia); Treatment 2: Phytopathogen control (PDA disc with phytopathogen mycelium); Treatment 3: Confrontation (*Trichoderma* spp. conidia + PDA disk with phytopathogen mycelium); and Treatment 4: Sterile distilled water.

### 2.6. Crude Extracellular Extract of Trichoderma spp.

TSMICH7 was the best strain that showed higher pathogen inhibitions in the in vivo test. For this reason, this *Trichoderma* strain was selected to obtain the extracellular extract and subsequently carry out the determinations of reducing sugars content and the lytic enzyme activities.

*Trichoderma* was grown in liquid to promote the secretion of antifungal compounds in the presence or absence of *N. parvum*, *C. gloeosporioides*, *Diaporthe* sp., and *P. perseae*. For growth in the absence of phytopathogens, an actively growing disc mycelium of Trichoderma spp. was inoculated in 100 mL of Czapek Dox culture medium and incubated at 28 °C at 120 rpm for 168 h. The same procedure was followed for the growth in the presence of phytopathogens, but 0.8 g of dry mycelium of each phytopathogen was added to the culture medium. The pre-inoculum of the phytopathogens was produced for 8 days at 28 °C with stirring at 120 rpm in 150 mL of potato dextrose broth to have enough mycelium.

At the end of this time, the mycelium was filtered and dried in the oven at a temperature of 50 °C for 48 h; then, it was kept at room temperature under aseptic conditions. The Czapek Dox culture was used as a blank without inoculation and was incubated under the same conditions. Obtaining crude extracellular extracts consisted of taking 10 mL of the medium every 24 h. The sample was filtered with Whatman #1 paper to separate the biomass from the supernatant and subsequently centrifuged (Heraeus Megafuge 16R, Thermo Fisher Scientific, Waltham, MA, USA) at 10,000 rpm for 8 min at 4 °C. The supernatant was kept at 4 °C until it was used to determine reducing sugars and enzymatic activities (chitinase and glucanase).

### 2.7. Reducing Sugars Content

The content of reducing sugars in each crude extracellular extract was determined by using the DNS method (3,5-dinitrosalicylic acid) [29]. In glass tubes, 0.5 mL of the sample and 0.5 mL of distilled water were added and homogenized. Then, 1 mL of DNS reagent (1 g of DNS and 30 g of sodium–potassium tartaric acid in 80 mL of 0.5 N NaOH at 45 °C) was added to this mixture. After dissolution, the tubes were immersed in boiling water and maintained for 5 min. The tubes were removed and cooled at room temperature for 15 min. Then, 2.5 mL of distilled water was added and vortex homogenized. The absorbance at 550 nm of the mixture in the tubes was measured using a microplate reader (FI-01620, Thermo Fisher Scientific), adding 300 μL of the mixtures from the tubes to each well. A glucose calibration curve (0–1.6 g/L) was used to report reducing sugars in g/L.

*2.8. Chitinase and Glucanase Activities*

Monitoring was carried out to confirm the antagonistic strain's possible secretion of compounds with antifungal properties. This process consisted of identifying the maximum point (in time) where the most significant amount of reducing sugars is present (DNS colorimetric test) and taking daily measurements. If the amount of reducing sugars increased during the confrontation, it was considered an indication that the lytic enzymes (chitinases and glucanases) act on the pathogen's cell walls, degrading the chitin polymers and glucan. Glucanase and chitinase activities were determined as the difference between the production activity of reducing sugars without substrate minus the enzymatic activity of reducing sugars with the substrate and from co-inoculations with their corresponding controls (single inoculations) [30].

2.8.1. Chitinase Activity

The procedure for analyzing chitinase activity using the specific shrimp shell chitin substrate was based on the methodology proposed by Kumar et al. [31] with modifications. An amount of 0.5 mL of crude extracellular extracts was taken, and 0.5 mL of shrimp shell chitin (0.5%) dissolved in McIlvaine's buffer (pH 4.0) was added to obtain a final volume of 1 mL. The mixture was homogenized and incubated at 40 °C for 40 min under agitation. At the end of this time, 1 mL of the DNS reagent was added to these tubes to determine reducing sugars. Chitinase activity was expressed as µg of reducing sugars released $min^{-1}$ $mL^{-1}$.

2.8.2. Glucanase Activity

Glucanase activity uses the specific substrate laminarin based on the protocol of Kumar, Amaresan, Bhagat, Madhuri, and Srivastava [31] with some modifications. From the crude extracellular extracts, 0.5 mL was taken and placed in glass tubes. An amount of 0.5 mL of laminarin (0.05 M) dissolved in citrate buffer (0.05 M pH 4.8) was added to these tubes, obtaining a final volume of 1 mL. The mixture was homogenized and incubated at 40 °C for 40 min under agitation. At the end of this time, 1 mL of the DNS reagent was added to these tubes to determine reducing sugars. Glucanase activity was expressed as µmol of glucose released $min^{-1}$ $mL^{-1}$.

*2.9. Statistical Analysis*

All experiments were performed in triplicate. Data were evaluated using analysis of variance (ANOVA), followed by the Least Significant Difference (LSD) test at a 95% confidence level, using the Statgraphics Centurion XVI program (StatPoint Technologies, Inc., Warrenton, VA, USA).

**3. Results**

*3.1. Isolation and Identification of Trichoderma spp. Strains*

Twenty-five native strains were isolated from the avocado cultivation orchards, principally from soil and root. More native strains of *Trichoderma* were isolated in Michoacan than in the state of Jalisco. The micro- and macroscopic characteristics of the colonies were considered to establish the identification with traditional techniques of *Trichoderma* spp. Regarding the microscopic characterization, all samples showed a similar shape (proliferating with conidiation predominantly effuse, appearing granular or powdery due to dense conidiation, rapidly turning yellowish-green to dark green, or producing tufts or pustules fringed by sterile white mycelium), size (6 cm in 96 h), phialides (ampulliform to lageniform, usually 3–4 verticillate, occasionally paired, mostly 3.5–7.5 × 2.5–3.8 µm, terminal phialides up to 10 µm long), and conidia (globose to obovoid, mostly 2.5–3.5 × 2.1–3.0 µm, smooth-walled, subhyaline to pale green).

Table 2 shows the traditional morphological macroscopic characterization. In general, the colonies were fast-growing (3–4 days), hyaline aerial mycelium, tangled, woolly, and varied in color, for example, from colorless to yellow, amber, opaque reddish, greenish-

yellow, and with a slight coconut smell depending on the species. The presence or not of diffusible pigments and concentric rings was observed. Thus, according to the micro- and macro-characterization, the samples belong to the genus *Trichoderma* spp. However, molecular identification is more accurate in determining both genus and species. Therefore, six strains (TSMICH7, TSMICH8, TRMICH9, TSMICH10, TSMICH15, and TRJAL25) were selected for molecular identification due to their highest ability to inhibit avocado pathogens.

**Table 2.** Traditional morphological macroscopic characteristics of native strains of *Trichoderma* spp. isolated from avocado orchards.

| Sample Code | Sampling Material | Morphological Characteristic | | | |
| --- | --- | --- | --- | --- | --- |
| | | Radial Growth | Mycelial Color | Presence of Diffusible Pigments | Concentric Rings |
| TSMICH1 | Soil | 4.1 cm in 120 h | Green | No | Yes |
| TSMICH2 | Soil | 4.8 cm in 120 h | Green | No | Yes |
| TSMICH3 | Root | 4.5 cm in 120 h | Green | Yes | No |
| TSMICH4 | Soil | 4.7 cm in 120 h | White | Yes | No |
| TSMICH5 | Root | 4.7 cm in 120 h | Green | Yes | Yes |
| TSMICH6 | Soil | 4.8 cm in 120 h | White-yellow | Yes | Yes |
| TSMICH7 | Root | 5 cm in 96 h | Green and white | Yes | Yes but undefined |
| TSMICH8 | Root | 4.9 cm in 96 h | White | No | Yes but undefined |
| TSMICH9 | Root | 4.8 cm in 96 h | White and beige | No | No |
| TSMICH10 | Soil | 4.8 cm in 96 h | Beige | Yes | Yes but undefined |
| TSMICH11 | Soil | 4.6 cm in 120 h | Green-beige | No | No |
| TSMICH12 | Soil | 4.8 cm in 120 h | Green | No | Yes but undefined |
| TSMICH13 | Soil | 4.8 cm in 120 h | White | Yes | No |
| TSMICH14 | Soil | 4.7 cm in 120 h | Beige | Yes | No |
| TSMICH15 | Soil | 4.8 cm in 96 h | Beige | Yes | No |
| TSMICH16 | Soil | 4.7 cm in 120 h | White | No | Yes |
| TSMICH17 | Soil | 4.6 cm in 120 h | Green | Yes | Yes |
| TSMICH18 | Soil | 4.6 cm in 120 h | White | Yes | Yes |
| TSMICH19 | Root | 4.5 cm in 120 h | Green and white | No | Yes but undefined |
| TSMICH20 | Soil | 4.7 cm in 120 h | Green | No | Yes |
| TSMICH21 | Soil | 4.5 cm in 120 h | Yellow | No | No |
| TSMICH22 | Soil | 4.7 cm in 120 h | Green and white | Yes, but little presence | Yes |
| TRJAL23 | Root | 4.9 cm in 120 h | Green | No | Yes |
| TRJAL24 | Root | 5 cm in 96 h | Green | No | Yes |
| TRJAL25 | Root | 4.9 cm in 120 h | Green | No | Yes |

The amplification of the genomic DNA of the six strains with the primers ITS1 and ITS4 and the *Tef 1-α* gene was approximately 600 base pairs (bp). The purified fragments were sequenced and compared for identity with the nucleotide sequences of reported fungi from the National Center Biotechnology Information (NCBI). All six strains were found to be closely related based on phylogenetic analysis of the aforementioned regions. In addition, they were grouped into subclades with sequences previously reported in GenBank of *Trichoderma harzianum* and were differentiated from the subclades of other *Trichoderma* species, suggesting that they belong to this genus (Figure 1, Table 3).

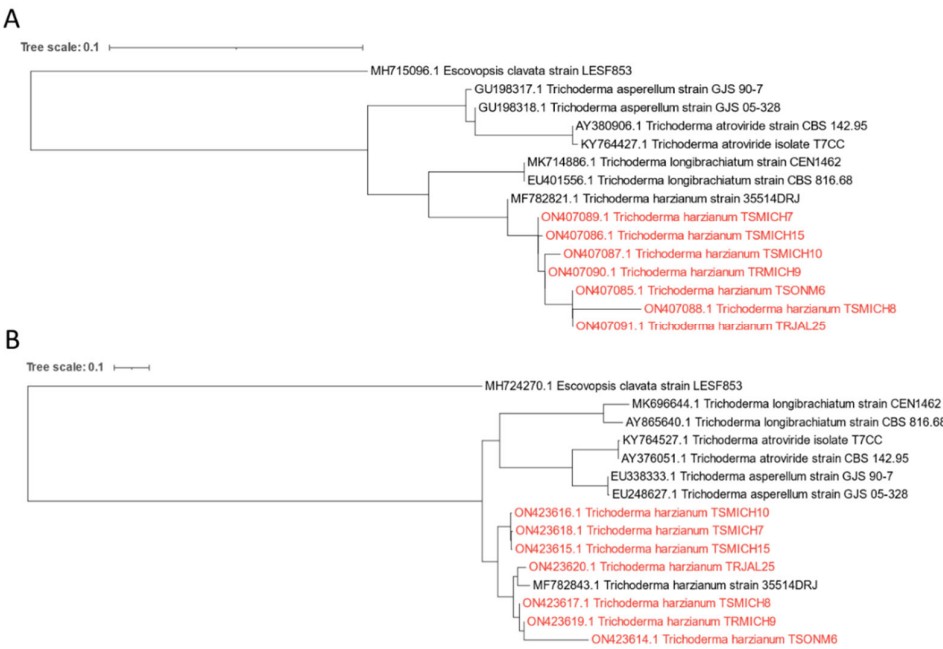

**Figure 1.** Phylogenetic trees based on ITS (**A**) and *Tef 1-α* (**B**) sequences of *Trichoderma* isolates.

**Table 3.** BLASTN results of *Trichoderma* spp. sequences.

| Strain | Best Hit | ITS Region | | Translation Elongation Factor *Tef 1-α* | |
| --- | --- | --- | --- | --- | --- |
| | | % Identity | GenBank Accession Number | % Identity | GenBank Accession Number |
| TSMICH7 | *Trichoderma harzianum* | 99.81 | ON407089 | 100 | ON423618 |
| TSMICH8 | *Trichoderma harzianum* | 97.70 | ON407088 | 99.31 | ON423617 |
| TSMICH9 | *Trichoderma harzianum* | 99.62 | ON407090 | 99.09 | ON423619 |
| TSMICH10 | *Trichoderma harzianum* | 98.90 | ON407087 | 99.65 | ON423616 |
| TSMICH15 | *Trichoderma harzianum* | 99.63 | ON407086 | 99.82 | ON423615 |
| TRJAL25 | *Trichoderma harzianum* | 98.56 | ON407091 | 98.65 | ON423620 |

*3.2. In Vitro Evaluation of the Antagonistic Capacity of Trichoderma spp.*

The antagonistic capacity of the 25 isolated strains of *Trichoderma* spp. against phytopathogenic fungi, *N. parvum*, *C. gloeosporioides*, *Diaporthe* sp., and *P. perseae* were analyzed in vitro. TSMICH7, TSMICH8, TRMICH9, TSMICH10, TSMICH15, TRJAL24, and TRJAL25 strains showed different behavior on phytopathogen inhibition, being superior to the rest of the native strains of *Trichoderma* spp. (Figure 2). These strains presented a marked antagonistic capacity, showing undifferentiated inhibition on the four phytopathogens studied.

The inhibition percentage of *Trichoderma* spp. (TSMICH7, TSMICH8, TRMICH9, TSMICH10, TSMICH15, TRJAL24, and TRJAL25 strains) against the pathogens was close to or higher than 80% in 96 h, presenting a growth rate of 4 mm per day after 12 h. The rest of the strains of *Trichoderma* spp. carried it out in 120 h, demonstrating that the growth by the antagonist was more significant than the pathogen. The statistical analysis presented a significant effect for the factors involved: strain, pathogen, and interaction ($p \leq 0.05$). Two inhibition forms were observed—one by forming inhibition halos (antibiosis), and the other by hyperparasitism, which is the overgrowth of the antagonist produced on the pathogenic strain. Of the 25 strains analyzed, 11 inhibited the growth of phytopathogens by competition, forming inhibition halos, while the remaining 14 had the capacity to hyperparasite the phytopathogens.

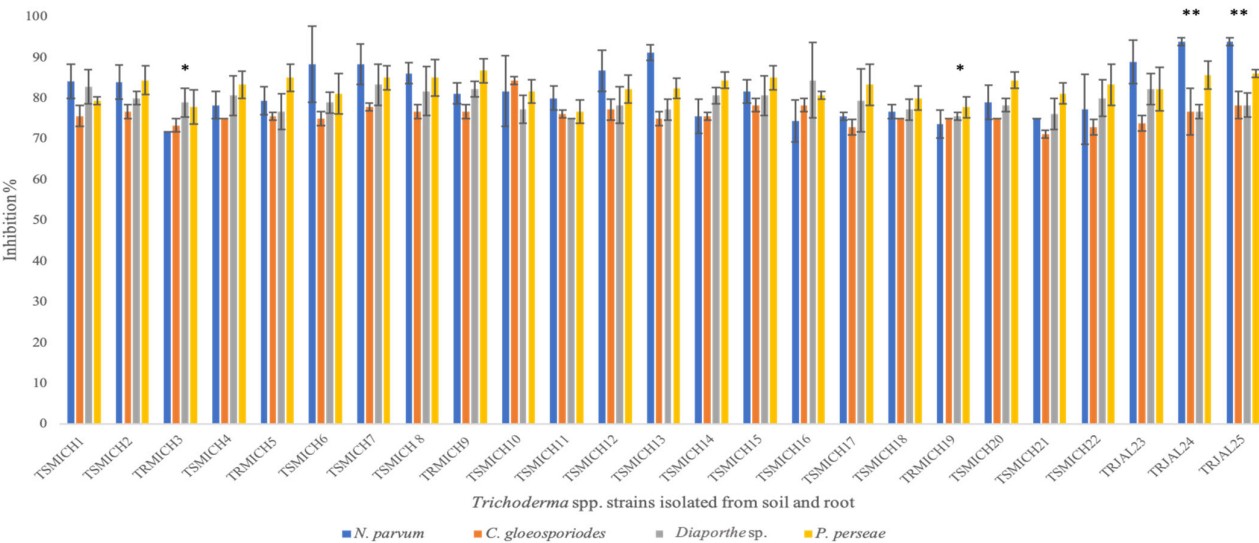

**Figure 2.** Growth inhibition of *N. parvum*, *C. gloeosporioides*, *Diaporthe* sp., and *P. perseae* for *Trichoderma* spp. strains isolated from soil and root of avocado crops from the region Jalisco and Michoacan in a period of 120 h. Means ± standard deviation (bar); * less interaction; ** higher interaction ($p < 0.05$).

The inhibition zone produced by the biocontroller against the pathogenic strain increased as time passed and was simultaneously accompanied by the destruction of the fungal mycelium obtained up to that moment. The isolates with the best antagonistic behavior were presented by TRJAL24 and TRJAL25 strains. However, these two samples did not have a significant difference, and TRJAL25 was chosen for further studies. Therefore, TRJAL25 was considered the sample with the highest antagonistic activity (93.9% inhibition for *N. parvum*, 78.3% inhibition for *C. gloeosporioides*, 78.3% *Diaporthe* sp., and 86.1% for *P. perseae*), and those that showed less antagonism were TRMICH3 and TRMICH19, reflecting an average antagonism of less than 80%. This result does not mean that they can be ruled out for use as biological control; they are also considered promising strains, as they show antagonistic activity higher than 70%.

The behavior of each phytopathogen was different, and the strains of *Trichoderma* spp. showed acceptable performance when confronted with phytopathogens. We conducted a morphometric study to elucidate the microscopical changes in the structure of both organisms when confronting them (Table 4). This analysis shows that after 96 h, the antagonistic strain undergoes a thickening of hyphae, while the phytopathogen tends to thin, except for *Diaporthe* sp., which tends to thicken. *Trichoderma's* behavior is possibly attributed to the secretion of secondary metabolites favoring the cell wall degradation of pathogens, turning them into a nutritional source for *Trichoderma*. According to the results of the analysis of variance, the factors of strain, pathogen, and interactions involved have statistically significant effects ($p \leq 0.05$).

*3.3. In Vivo Evaluation of the Antagonistic Capacity of Trichoderma spp.*

For the in vivo analysis, only six isolates identified as *T. harzianum* were considered in this study (TSMICH7, TSMICH8, TRMICH9, TSMICH10, TSMICH15, and TRJAL25). The in vivo confrontations were tested for seven days (Figure 3). In this study, we observed that when confronting the native strains of *T. harzianum* against *N. parvum*, the fruits showed the development of soft rot and accelerated maturation, increasing the colonization of the pulp. Even when the phytopathogen showed aggressiveness, leaving the evidence mentioned above, the strains of *T. harzianum* showed an antagonistic effect in inhibiting *N. parvum*. However, this pathogen is more resistant to *T. harzianum* than other pathogens.

**Table 4.** Morphometric analysis of the hyphae of the antagonistic strains (*Trichoderma* spp.) and the pathogenic ones and their confrontations at 96 h.

| Antagonistic Strains *Trichoderma* | Diameter of Hyphae (µm) | Pathogen | Diameter of Hyphae of the Phytopathogen (µm) | Confrontation *Trichoderma*/Phytopathogen Diameter of Hyphae (µm) | | Diameter Differential after Confrontation (µm) * | |
|---|---|---|---|---|---|---|---|
| | | | | *Trichoderma* | Phytopathogen | *Trichoderma* | Phytopathogen |
| TSMICH7 | 2.85 ± 0.49 [b] | *N. parvum* | 4.92 ± 1.41 [ab] | 4.34 ± 0.59 [b] | 4.95 ± 0.74 [a] | +1.49 | +0.03 |
| | | *C. gloeosporioides* | 5.96 ± 1.06 [a] | 4.50 ± 0.44 [b] | 4.36 ± 0.69 [ab] | +1.65 | −1.60 |
| | | *Diaporthe* sp. | 3.58 ± 0.53 [c] | 4.95 ± 1.40 [b] | 3.59 ± 0.59 [b] | +2.10 | +0.002 |
| | | *P. perseae* | 5.62 ± 2.08 [a] | 3.90 ± 0.41 [c] | 3.89 ± 0.42 [b] | +1.05 | −1.73 |
| TSMICH8 | 2.08 ± 0.81 [c] | *N. parvum* | 4.92 ± 1.41 [ab] | 4.97 ± 1.05 [b] | 3.80 ± 0.69 [b] | +2.89 | −1.11 |
| | | *C. gloeosporioides* | 5.96 ± 1.06 [a] | 5.72 ± 1.44 [a] | 3.49 ± 0.92 [bc] | +3.63 | −2.46 *** |
| | | *Diaporthe* sp. | 3.58 ± 0.53 [c] | 3.47 ± 0.77 [bc] | 3.88 ± 1.24 [b] | +1.38 | +0.29 |
| | | *P. perseae* | 5.62 ± 2.08 [a] | 3.54 ± 0.92 [c] | 4.45 ± 0.83 [a] | +1.45 | −1.17 |
| TSMICH9 | 3.34 ± 0.59 [a] | *N. parvum* | 4.92 ± 1.41 [ab] | 5.17 ± 1.16 [ab] | 3.67 ± 1.21 [b] | +1.80 | −1.24 |
| | | *C. gloeosporioides* | 5.96 ± 1.06 [a] | 4.44 ± 1.23 [b] | 3.55 ± 0.67 [b] | +1.10 | −2.40 |
| | | *Diaporthe* sp. | 3.58 ± 0.53 [c] | 4.91 ± 0.38 [b] | 4.43 ± 1.37 [a] | +1.56 | +0.85 |
| | | *P. perseae* | 5.62 ± 2.08 [a] | 4.63 ± 0.52 [b] | 3.92 ± 0.68 [b] | +1.29 | −1.70 |
| TSMICH10 | 3.48 ± 0.20 [a] | *N. parvum* | 4.92 ± 1.41 [ab] | 3.78 ± 0.72 [bc] | 4.63 ± 0.65 [a] | +0.29 | −0.28 |
| | | *C. gloeosporioides* | 5.96 ± 1.06 [a] | 2.95 ± 0.32 [cd] | 3.21 ± 0.71 [bc] | −0.52 | −2.74 |
| | | *Diaporthe* sp. | 3.58 ± 0.53 [c] | 3.24 ± 0.59 [c] | 2.69 ± 0.49 [c] | −0.24 | −0.88 |
| | | *P. perseae* | 5.62 ± 2.08 [a] | 2.89 ± 0.30 [d] | 2.58 ± 0.70 [c] | −0.59 | −3.04 ** |
| TSMICH15 | 3.56 ± 0.86 [a] | *N. parvum* | 4.92 ± 1.41 [ab] | 4.19 ± 0.45 [b] | 4.20 ± 1.34 [ab] | +0.63 | −0.71 |
| | | *C. gloeosporioides* | 5.96 ± 1.06 [a] | 3.24 ± 0.60 [c] | 3.42 ± 0.96 [bc] | −0.31 | −2.53 |
| | | *Diaporthe* sp. | 3.58 ± 0.53 [c] | 4.99 ± 0.68 [b] | 4.69 ± 0.82 [a] | +1.43 | +1.10 |
| | | *P. perseae* | 5.62 ± 2.08 [a] | 5.90 ± 1.53 [a] | 2.58 ± 0.70 [c] | +2.34 | −3.04 |
| TRJAL25 | 2.97 ± 0.24 [ab] | *N. parvum* | 4.92 ± 1.41 [ab] | 4.32 ± 0.76 [b] | 4.34 ± 0.77 [ab] | +2.97 | −0.57 |
| | | *C. gloeosporioides* | 5.96 ± 1.06 [a] | 2.28 ± 0.42 [d] | 3.13 ± 0.72 [bc] | −0.69 | −2.83 |
| | | *Diaporthe* sp. | 3.58 ± 0.53 [c] | 3.92 ± 0.43 [bc] | 4.43 ± 0.64 [a] | +0.94 | +0.84 |
| | | *P. perseae* | 5.62 ± 2.08 [a] | 4.35 ± 0.69 [b] | 2.58 ± 0.70 [c] | +1.37 | −3.04 |

* Positive value (+) indicates hyphal thickening, negative value (−) indicates thinning of the hyphae. Different letters in each column indicate significant differences between antagonistic strains ($p < 0.05$). ** Less interaction; *** higher interaction.

Based on the proposed scale (Figure S1), the TSMICH7 strain presented an inhibition of 63.3% against *N. parvum* (causal agent of stem rot) and the rest of the strains showed percentages lower than 50%. Concerning the confrontation of the isolated strains of *T. harzianum* vs. *C. gloeosporioides*, the antagonistic strains presented more than 90% inhibition of this pathogen, while TSMICH8 and TRJAL25 strains were highlighted with 100% inhibition. The consistency was maintained until the evaluation; the same behavior was observed with the phytopathogens *Diaporthe* sp. and *P. persea*. For the inhibition effect of *Diaporthe* sp., the best antagonistic strains were TSMICH7 and TSMICH15, with 90% and 93% inhibition, respectively; the rest of the strains showed inhibition of 70% to 73%. Finally, for the inhibition of *P. perseae*, the strains that showed the highest percentage of inhibition (94%) were TSMICH7 and TSMICH15.

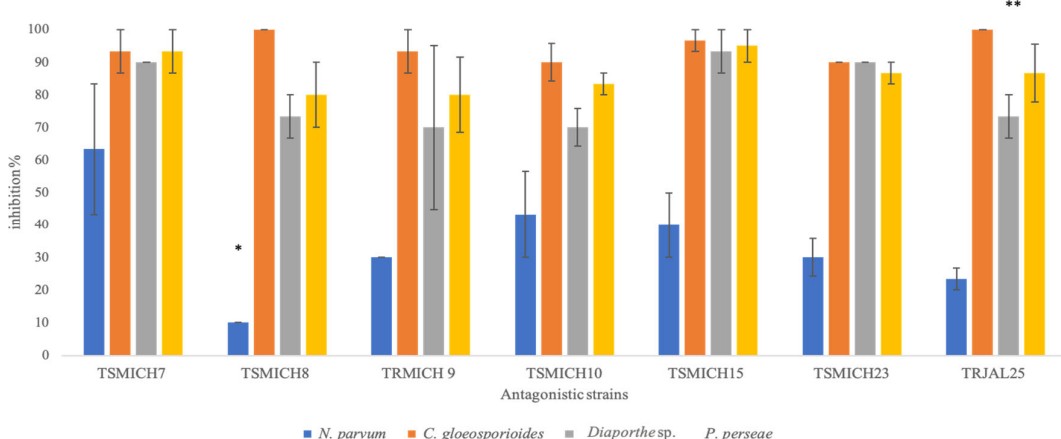

**Figure 3.** Inhibition percentages from the in vivo confrontation of antagonistic strains against phytopathogens. Data plotted as means ± standard error. * Less interaction; ** higher interaction ($p < 0.05$).

The rotting damage in control fruits caused by the phytopathogens and antagonists is evidenced in Figure 4 (scale in the percentage of rotting of the fruit body and peduncular area), where exciting results can be observed when comparing the pathogenic variability caused. The necrotic area in the fruit caused by the inoculation of the phytopathogen *N. parvum* reached 83.3%. On the other hand, *C. gloeosporiodes* showed 30% rot. The damage caused by *Diaporthe* sp. was relatively low (23.3%) compared to *P. perseae* (66.7%). *C. gloeosporioides*, *Diaporthe* sp., and *P. perseae* inoculated at physiological maturity showed very little development of the pathogen.

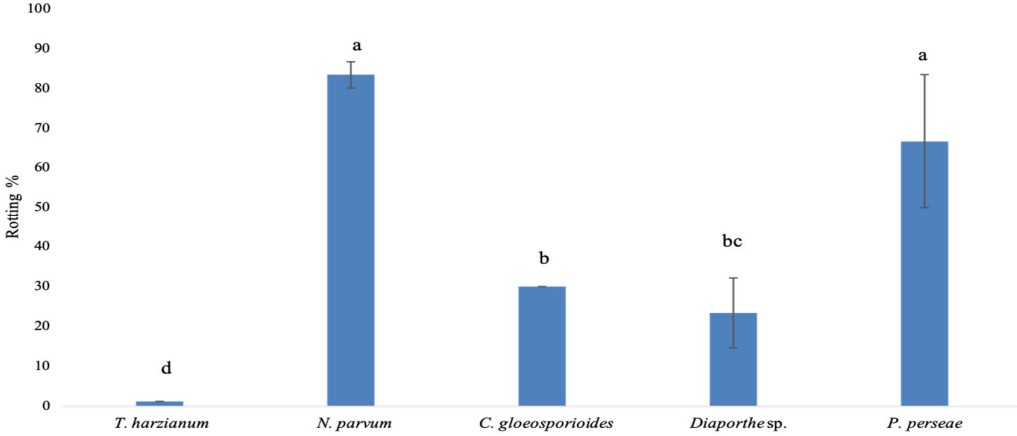

**Figure 4.** Percentages of rot in control fruits from in vivo test of *T. harzianum* and phytopathogen strains of post-harvest avocado fruit. Data plotted as means ± standard error. Different letters indicate significant differences ($p < 0.05$).

According to the results of the analysis of variance, the factors of strain, pathogen, and interactions involved have significant effects ($p \leq 0.05$). Regarding the control fruits inoculated with *Trichoderma* spp., the fruits did not show signs of rotting, remaining intact until the time of the evaluation. The isolates of *Trichoderma* spp. against phytopathogens were conducted through bioassays under controlled conditions. The results show a significant effect in the development reduction of *C. gloeosporioides*, *Diaporthe* sp., and *P. persea* in the post-harvest fruit inoculated with the organisms described above, principally *T. harzianum* (TSMICH7).

### 3.4. Lytic Activity (Chitinase and Glucanase) of Trichoderma harzianum

The best strain in the in vivo analysis was *T. harzianum* (TSMICH7). Hence, this strain was used to determine the reducing sugars and lytic activities (chitinase and glucanase).

#### 3.4.1. Reducing Sugars

Determining enzymatic activity allows identifying the maximum point in the production of reducing sugars hypothetically induced by the action of the hydrolytic enzymes of interest (glucanases and chitinases). The reducing sugars quantification of the antagonistic and non-antagonistic complex in basal medium (crude extracellular extract) revealed the highest amount between 72 and 120 h (Figure 5). For the strain, *T. harzianum* (TSMICH7) was sufficient at 24 h to produce the maximum amount of 13.6 g/L of reducing sugars for this strain.

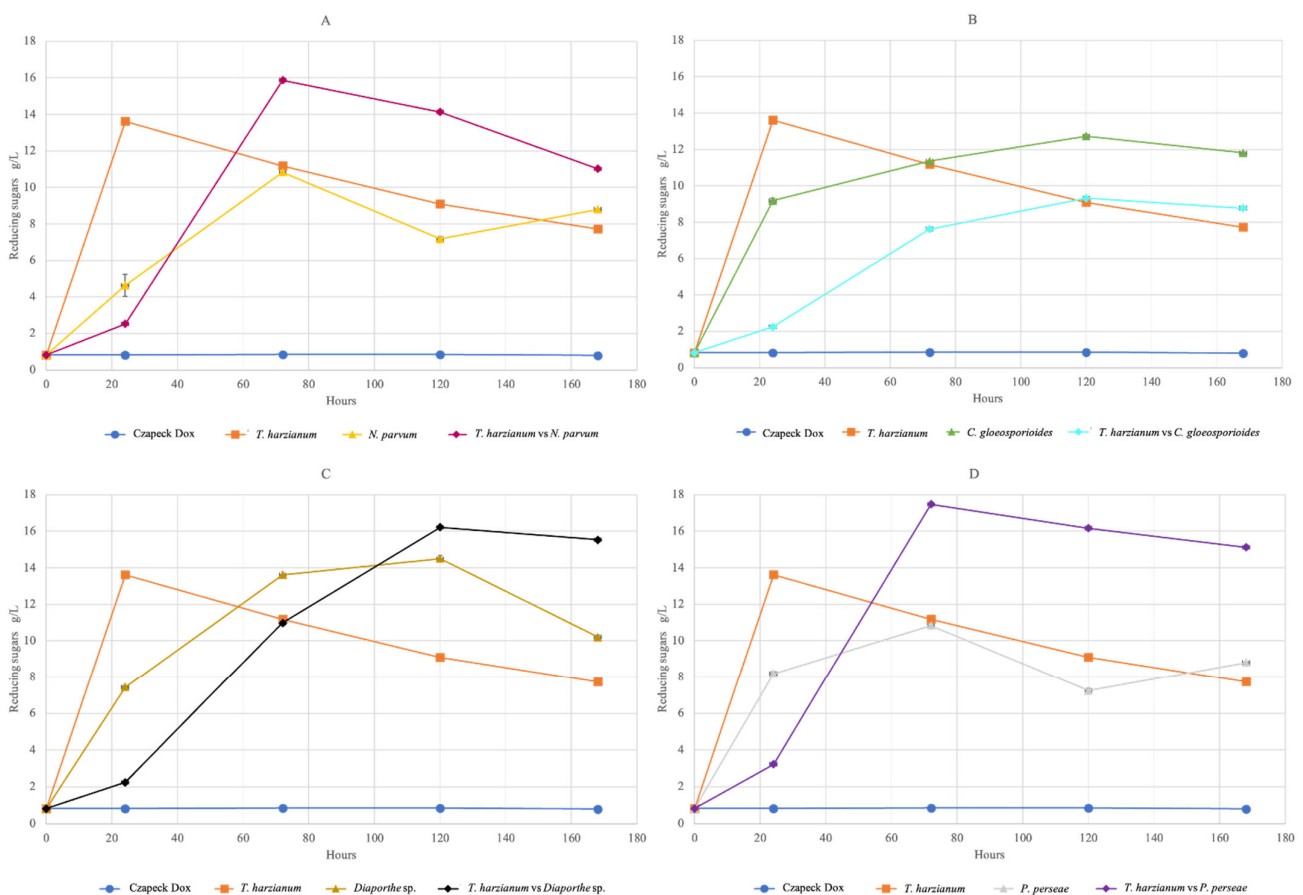

**Figure 5.** Contents of reducing sugars in antagonistic and non-antagonistic strains with their respective complexes at different times. (**A**) Assay with *N. parvum*; (**B**) assay with *C. gloeosporioides*; (**C**) assay with *Diaporthe* sp.; (**D**) assay with *P. perseae*. Mean ± standard deviation (bar).

When carrying out the confrontation of *T. harzianum* with each phytopathogen (*N. parvum, C. gloeosporioides, Diaporthe* sp., and *P. perseae*), it was possible to record that *T. harzianum* vs. *N. parvum* reached 15.8 g/L (Figure 5A) and *T. harzianum* vs. *P. perseae* produced 17.4 g/L (Figure 5D) in a period of 72 h. In comparison, the complex *T. harzianum* vs. *C. gloeosporioides* reflected 9.3 g/L (Figure 5B), and *T. harzianum* vs. *Diaporthe* sp. reached an amount of 16.2 g/L (Figure 5C) in 120 h. The phytopathogens exhibited a similar pattern in terms of the durations (72–120 h) of increased reducing sugar production, but they varied in the levels of these sugars, ranging between 7 and 15 g/L. Finally, the control used (Czapeck Dox medium) was not more than 8 g/L of reducing sugars constantly during

the identification period. The factors (strain, pathogen, and interaction) involved have significant effects ($p \leq 0.05$).

### 3.4.2. Total Enzyme Activity of Chitinase and Glucanase

In this study, the determination of the lytic activity of both chitinase and glucanase reflected varied data, without any specific pattern that obeys the behavior of the organisms tested and without compromising the mechanisms evidenced by the antagonist (*T. harzianum*). The total enzymatic activity for chitinase revealed the amount of 1522.47 µg reducing sugars $min^{-1}mL^{-1}$ in only 24 h for *T. harzianum* (TSMICH7); in the following hours, this amount gradually decreased (Figure 6). For glucanase, the total enzymatic activity was 465.47 µmol glucose released $min^{-1} mL^{-1}$ at 168 h (Figure 7).

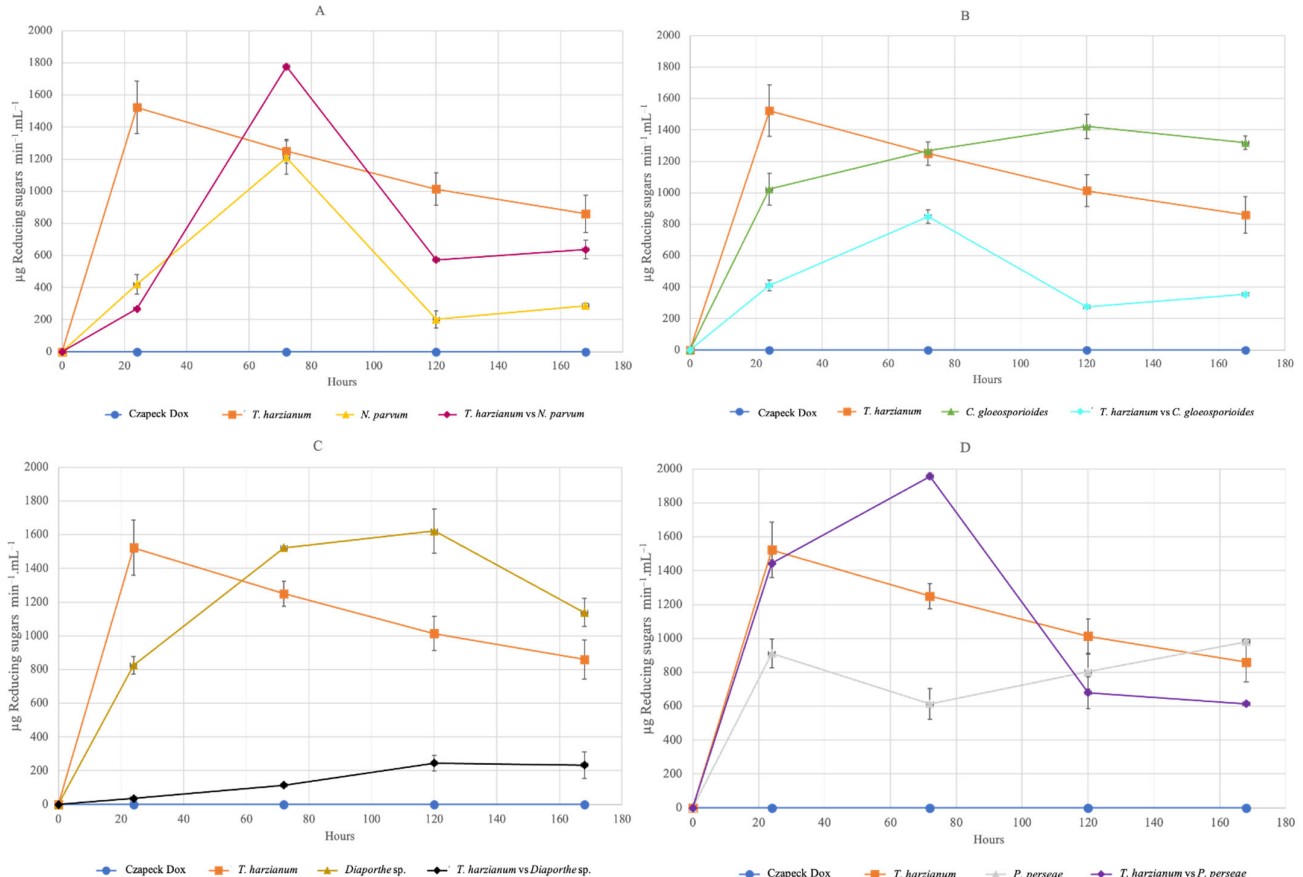

**Figure 6.** Total chitinase enzyme activity of native TSMICH7 strain (*T. harzianum*) against phytopathogens evaluated at 0, 24, 72, 120, and 168 h. (**A**) Assay with *N. parvum*; (**B**) assay with *C. gloeosporioides*; (**C**) assay with *Diaporthe* sp.; (**D**) assay with *P. perseae*. Mean ± standard deviation (bar).

Regarding the total chitinase enzymatic activity of the antagonistic and non-antagonistic complexes confronted in the same way, they express different quantities—*T. harzianum* with the phytopathogen *N. parvum* (Figure 6A) and *P. perseae* (Figure 6D) expressed amounts of 1779.72 and 1956.74 µg reducing sugars $min^{-1} mL^{-1}$, respectively, higher than those produced individually by the isolated *T. harzianum* for *C. gloeosporioides*, with 848.17 µg reducing sugars $min^{-1} mL^{-1}$ (Figure 6B) in a period of 72 h and at 120 h. *T. harzianum* vs. *Diaporthe* sp. expressed 244.92 µg reducing sugars $min^{-1} mL^{-1}$ (Figure 6C). The total enzymatic activity for chitinase of the phytopathogens was 1209.80, 1422.98, 1621.96, and 980.82 µg reducing sugars $min^{-1} mL^{-1}$ in 120 h for *N. parvum*, *C. gloeosporioides*, *Diaporthe* sp., and *P. perseae*, respectively.

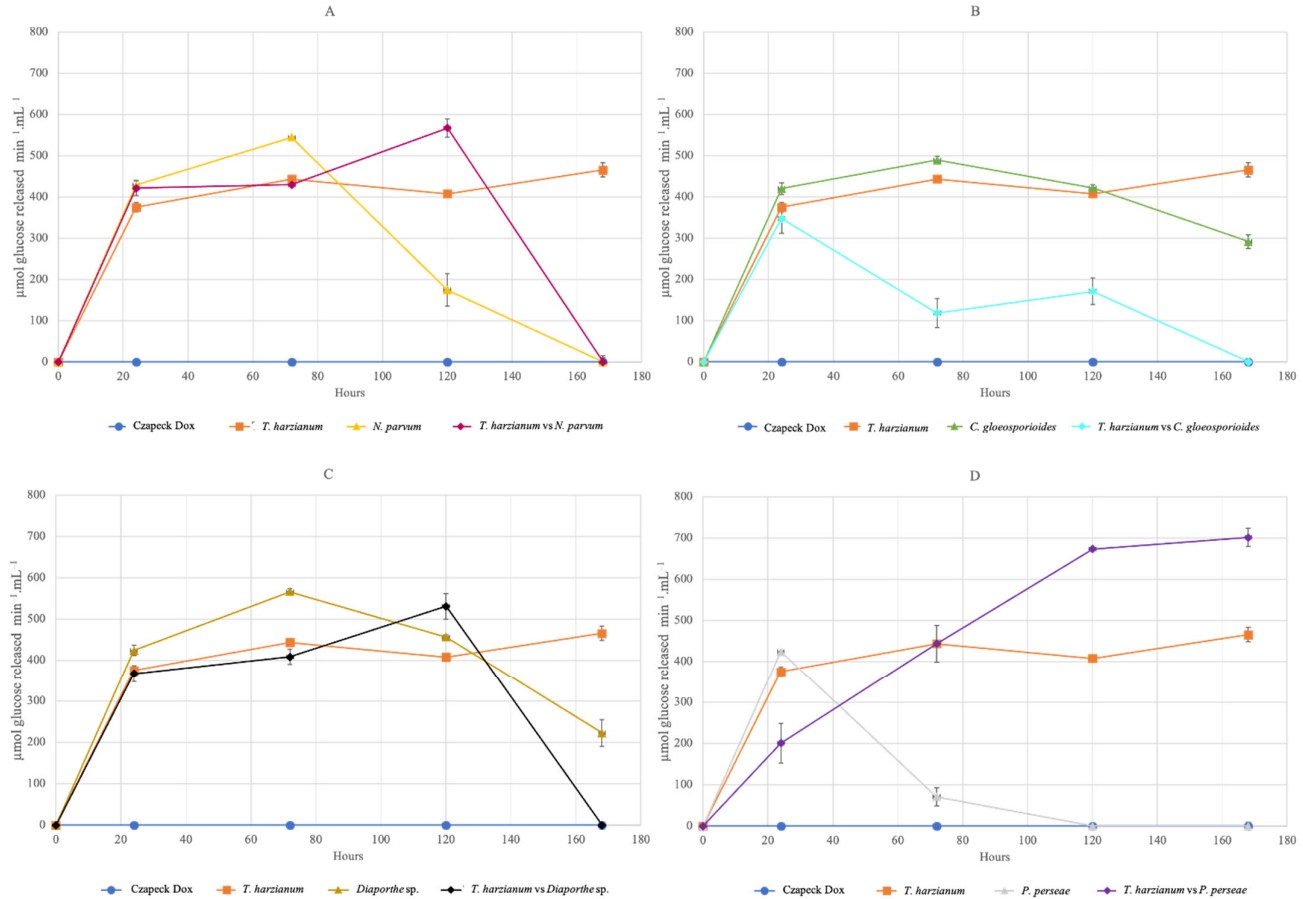

**Figure 7.** Total glucanase enzyme activity of native TSMICH7 strain (*T. harzianum*) against phytopathogens evaluated at 0, 24, 72, 120, and 168 h. (**A**) Assay with *N. parvum*; (**B**) assay with *C. gloeosporioides*; (**C**) assay with *Diaporthe* sp.; (**D**) assay with *P. perseae*. Mean ± standard deviation (bar).

Concerning the total enzymatic activity for glucanase in the organisms, the first instance was the diversification in times taken to obtain higher production of released glucose. The complex *T. harzianum* co-inoculated with *P. perseae* (Figure 7D) produced the highest amount of glucose released (701.47 $\mu$mol min$^{-1}$mL$^{-1}$) in 168 h. At the same time, the rest of the confrontations ranged from 300 to 550 $\mu$mol of glucose released min$^{-1}$mL$^{-1}$ in a time, which fluctuated from 24 to 120 h (Figure 7A–C). It is important to emphasize that three of the four confronted complexes are higher than the amount of glucose released for the amount produced by the isolated *T. harxianum* (465.47 $\mu$mol min$^{-1}$mL$^{-1}$) inoculated independently.

According to the determination of total enzymatic activity for glucanase in the phytopathogens, only a time range between 24 and 72 h was experienced for the highest activity. For *P. perseae*, 427.97 $\mu$mol glucose released min$^{-1}$mL$^{-1}$ was obtained in 24 h. In 72 h, *N. parvum* obtained 544.47, *C. gloeosporioides* obtained 488.97, and *Diaporthe* sp. obtained 565.97 $\mu$mol glucose released min$^{-1}$mL$^{-1}$. It is essential to highlight that the *T. harzianum* isolates co-inoculated with *N. parvum*, *C. gloeosporioides*, and *Diaporthe* sp. showed enzymatic activity at 120 h. All the factors evaluated showed a statistically significant effect on total enzymatic activity with a 95.0% confidence level for both chitinase and glucanase.

This study has also shown the activity of phytopathogenic organisms (*N. parvum*, *C. gloeosporioides*, *Diaporthe* sp., and *P. perseae*) in the production of reducing sugars. It was expected that they would not present high levels, since being inoculated individually and not being in front of an organism that produces hydrolytic secretions means their activity dynamics would be null, but it was not a general rule considering that the fun-

gal pathogens studied were also organisms. Therefore, they present the production of phytotoxic substances.

## 4. Discussion

### 4.1. Isolation and Identification of Trichoderma spp. Strains

*Trichoderma* was easily isolated from soil and root by conventional methods, primarily due to its rapid growth, abundant conidiation, chlamydospore formation, and colonization of organic substrates, which facilitate rapid development on various substrates [32]. The results obtained in this study agree with those reported by different authors when isolating *Trichoderma* from soil and roots from other crops or soils, for example, in avocado from Puebla, Mexico [33], in strawberry from Paraguay [34], in orange from Argentina [8], in apple from China [9], in banana from Malaysia [10], and in ferralitic soils in Cuba [34], showing that *Trichoderma* spp. is a fungus found in many types of environments around the world [35].

The characteristics of the fungal organism observed in the micro- and macroscopic studies are consistent with the genus of *Trichoderma* sp. Some authors also observed the same characteristics [33,36–39]. Due to the identity values and the subclades obtained in the phylogenetic analysis of the sequence alignments, it is verified that DNA barcoding markers such as the ITS region and *Tef 1-α* gene are reliable and reproducible techniques for identifying *T. harzianum* [40–42]. The isolation of *Trichoderma* spp. was greater in the orchards of the Michoacan state (22 strains) than in the orchards Jalisco state (3 strains), presuming that the edaphoclimatic characteristics, such as temperature and pH [43,44], are different in both states, even though these qualities are presented in the same geographic region. Additionally, Michoacan was sampled in autumn–winter, having the most suitable conditions for avocado cultivation, while in Jalisco, it was sampled in spring–summer, thus possibly affecting the occurrence of *Trichoderma* sp.

Another factor that alludes to the low or high incidence of *Trichoderma* sp. in both regions (Jalisco and Michoacan) could be attributed to the modernization with which the crop is managed to achieve increased productivity and fruit quality, referring to the irrational use of chemical products. This was also ratified by Komarek et al. [45]. Furthermore, modern agriculture has become dependent on chemical pesticides to control phytopathogenic organisms, which has caused pest resistance, changes in soil microbial diversity, and environmental pollution [46–48]. Microorganisms could be identified as *T. harzianum* based on their morphological characteristics and molecular techniques, as demonstrated by other researchers [40,41,49–51].

### 4.2. In Vitro Evaluation of the Antagonistic Capacity of Trichoderma spp.

The high antagonistic capacity of the six native strains (TSMICH7, TSMICH8, TR-MICH9, TSMICH10, TSMICH15, and TRJAL25) was close to or greater than 80% of inhibition. Similar results were reported by Hermosa et al. [52] and Sabbagh et al. [53], who reported that above 80% inhibition is acceptable in the biocontrol of pathogens that are transmitted by soil and roots. The variability of the *Trichoderma* strains' antagonistic capacity might be because each strain has a different mechanism(s) of action and growth rate, among other causes [41]. The forms of inhibition evidenced mainly in this research were mycoparasitism, antibiosis, or competition for food (substrate) and space, as suggested by [19,54–57]. The growth of the antagonist reflected a typical exponential form, reaching its stationary phase on day 4, which confirms the ability of this microorganism to adapt to in vitro growth conditions and rapidly colonize a given space, which favored the inhibition of the growth of phytopathogenic fungi, a situation that was demonstrated in the first hours of the confrontation.

Likewise, in the morphometric analysis for each of the isolated strains, we examined the effects produced on the phytopathogen by placing them in an in vitro confrontation situation, in which the damage caused by the antagonistic strain was clearly observed by reducing the growth of the phytopathogen. Mukherjee et al. [58] and Safari Motlagh

and Samimi [59] reported that the pathogen growth is affected by contact with *Trichoderma* and that some species attack directly by producing lysis of the mycelium, so *Trichoderma* can decrease or increase its attack, establishing a direct correlation between time and aggressiveness of the antagonist towards the pathogen, with greater aggressiveness in a shorter time.

The bioassays conducted with *Trichoderma* spp. have been effective, and by its nature, biological control does not eliminate but instead reduces the pathogen populations and, consequently, reduces the incidence of the disease [60]. Some studies have shown the effectiveness of *Trichoderma* spp. against *Botrytis cinerea* in strawberry [7]; *Alternaria alternata*, *Colletotrichum gloeosporioides*, and *Penicillium digitatum* in orange [8]; *Fusarium proliferatum* in apple [9]; *Fusarium oxysporum* in banana [10]; *Colletotrichum* sp. and *Fusarium* sp. in mangrove [11]; and *Fusarium incarnatum* in muskmelon [12], among others. However, despite the potential of these autochthonous antagonists, they must be further evaluated, clarifying the mechanisms through which they exert their action to improve their effectiveness when they are applied as microbial inoculants [61].

### 4.3. In Vivo Evaluation of the Antagonistic Capacity of Trichoderma spp.

The phytopathogen *N. parvum* caused greater relevance in this study due to the high colonization of the pulp and accelerated maturation in the post-harvest avocado fruit. Similar results regarding susceptibility were obtained in a study using blueberry [62] and grapevine [63]. In the rest of the inoculated avocados, the development of phytopathogens was counteracted by the antagonistic strain, which decreased susceptibility in the post-harvest fruit, allowing it to maintain its consistency. These results are possible due to the content of phenols in the exocarp of the fruits, since it plays an essential role in the fruit's defense mechanisms against attack by fungal pathogens [64] or in producing other excreted compounds as lytic enzymes by *T. harzianum* [19]. Likewise, this disease in the unripe fruits acts as an inactive infection and can manifest symptoms during or after the ripening process [65–67].

*T. harzianum* isolates against phytopathogens showed a significant effect in the reduction of development of *C. gloeosporioides*, *Diaporthe* sp., and *P. perseae*. Of the few studies available on this theme, it is found that the application of *T. harzianum* increased the firmness of the fruits by 1.2 times compared to the control plants. This behavior can be attributed to the induction of the biosynthesis of the phytohormone by activating plant defense mechanisms, mainly ethylene, responsible for the expression of plant maturation genes [68]. According to Colla et al. [69], firm fruits are more resistant to attack by microorganisms. The degree of damage caused by fungal pathogens in post-harvest fruits was appreciated with different behaviors, which was related to a dependence on extrinsic factors such as exposure time, temperature, and humidity; coupled with these factors, the corresponding life cycle of each pathogen was found [43].

### 4.4. Lytic Activity (Chitinase and Glucanase) of Trichoderma harzianum

The production of reducing sugars hypothetically is induced by the action of the hydrolytic enzymes (glucanases and chitinases). In this study, each antagonistic and non-antagonistic complex presented different capacities in the production of those metabolites. Those could be because *Trichoderma* has compounds that limit the growth of the pathogenic fungus when excreted into the environment. This type of antagonism, recognized as antibiosis, is a mode of action that could be associated with an extracellular chitinase, which causes the release of some oligomers from the fungus cell wall and induces the expression of mycoparasite endochitinases [70–74]. When these are released, they are spread and cause *Trichoderma* to begin infection on the target fungus before physical contact between them [75]. It is essential to highlight that the lytic capacity developed by *Trichoderma* depends more on the strain and the host than on the different species [55], as corroborated in our study.

*Trichoderma* strains differ in the expression levels of hydrolytic enzymes, which determines their antagonistic characteristics [76,77]. During mycoparasitism, *Trichoderma* could secrete not only enzymes such as chitinases and glucanases, but also proteases that hydrolyze the fungal cell wall. The total enzymatic activity for chitinase was at 24 h when it presented the highest production, while for glucanase, it was up to 168 h. The same pattern was evidenced when co-inoculating *T. harzianum* with each of the phytopathogens tested, so the variability in the lytic activity was attributed to the presence of extracellular enzymes of the antagonist with lytic activity, such as chitinases, glucanases or proteases, among others, on the hyphae of pathogens [78,79]. Likewise, Yan et al. [80] and Ting and Chai [30] reported that chitinase activity correlates directly with the microstructure and proportion of chitin in the cell wall of the fungi with which it interacts, which explains the importance of these enzymes in the antagonistic capacity of *Trichoderma* and therefore in its usefulness as a biopesticide.

## 5. Conclusions

The isolation and identification at the genus level (*Trichoderma*) of 25 native isolates were achieved in avocado crops. Six strains were identified at the species level by molecular techniques; five were from Michoacan from root and soil, and one was from Jalisco isolated from the root, resulting in *T. harzianum* for all strains. The six isolated strains of *T. harzianum* soil and root (TSMICH7, TSMICH8, TRMICH9, TSMICH10, TSMICH15, and TRJAL25) presented the greatest in vitro antagonistic capacity against the four phytopathogens tested, suggesting more than 80% inhibition and requiring 96 h to contact the target organism. The TRJAL25 strain was superior to all strains, showing 93.9% inhibition for *N. parvum*, 78.3% for *C. gloeosporioides*, 78.3% for *Diaporthe* sp., and 86.1% for *P. perseae*.

Two mechanisms of action by the *Trichoderma* strains were evidenced: the formation of inhibition halos and hyperparasitism. The morphological behavior in the confrontation (antagonist–phytopathogen complex) was diverse; notably, in its majority, the diameter of the phytopathogen hyphae decreased. The six strains of *T. harzianum* tested maintained considerable in vivo inhibition against phytopathogens (*C. gloeosporioides*, *Diaporthe* sp., and *P. perseae*), and that which best inhibited the four phytopathogens was the TSMICH7 (*T. harzianum*) strain.

*T. harzianum* proved to be more effective against the pathogens *P. perseae* and *N. parvum* at 72 h and *Diaporthe* sp. at 120 h, releasing the most significant amount of reducing sugars and the highest activity of chitinases. However, with these same three pathogens, the glucanase activity was higher at 120 h. Therefore, *T. harzianum* could be used as a biological control for inhibiting avocado pathogens such as *P. perseae*, *N. parvum*, and *Diaporthe* sp. By controlling these diseases, the quality and export of these fruits is more assured, consequently bringing a better economy to both farmers and the producing country.

**Supplementary Materials:** The following are available online at https://www.mdpi.com/xxx/s1, Figure S1: Percentage scale for fruit body rot (A) and stem rot (B) of avocado. The image was taken from López-López et al. [28].

**Author Contributions:** Conceptualization, J.A.A.-L. and O.M.-C.; formal analysis, J.P.G.-G.; methodology, S.O.-A. and M.P.-J.; validation, M.A.R.-G., M.G.Á.-N. and P.J.G.-M.; investigation and writing—original draft preparation, M.E.L.-L.; writing—review and editing, A.T.B.-M.; supervision, C.L.D.-T.-S. and M.G.-L. All authors have read and agreed to the published version of the manuscript.

**Funding:** This research received no external funding.

**Institutional Review Board Statement:** Not applicable.

**Informed Consent Statement:** Not applicable.

**Data Availability Statement:** The complete sequences of TSMICH7, TSMICH8, TSMICH9, TSMICH10, TSMICH15, and TRJAL25 (ITS region and translation factor *Tef 1-α*) have been deposited in the GenBank database under the accession numbers ON407086–ON407091 (ITS region) and ON423615–ON423620 (elongation factor *Tef 1-α*).

**Conflicts of Interest:** The authors declare no conflict of interest.

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
