# Peer review of "Isolation and Characterization of Trichoderma spp. for Antagonistic Activity against Avocado (Persea americana Mill) Fruit Pathogens"

_horticulturae, doi:10.3390/horticulturae8080714_

Round 1

Reviewer 1 Report

This research is interesting, may serve as a basis for the development and utilization of new formulations with Trichoderma spp. for antagonistic activity against avocado fruit pathogens. Only a minor revision is needed. Questions and suggestions are visible in the attached revised file.

Author Response

Thanks to the reviewer for all appropriate comments and suggestions. We attended all the observations according to each one of the comments.

Point 1. This research is interesting, may serve as a basis for the development and utilization of new formulations with Trichoderma spp. for antagonistic activity against avocado fruit pathogens. Only a minor revision is needed. Questions and suggestions are visible in the attached revised file.

Response 1.  All recommendations were attended.

Reviewer 2 Report

There are some major points need to be addressed:

1. There are many mistake spelling especially scientific name of fungi.

2. To identify Trichoderma into species level authors need to construct phylogenetic tree of combined sequences (tef1-alpha + ITS), only blasn search data is not sufficient.

3. Discussion part is look like review article, so it need to be revised and rearrangement, make it short and discuss point by point. 

Some minor points are in pdf. file

Author Response

Thanks to the reviewer for all appropriate comments and suggestions. We attended all the observations according to each one of the comments.

Point 1. There are many mistake spelling especially scientific name of fungi.

Response 1. All these errors were reviewed and corrected. The English form was sent to be reviewed by experts from the MDPI publisher.

Point 2. To identify Trichoderma into species level authors need to construct phylogenetic tree of combined sequences (tef1-alpha + ITS), only blasn search data is not sufficient.

Response 2. The phylogenetic tree was constructed, and it is presented as Figure 1.

Point 3. Discussion part is look like review article, so it need to be revised and rearrangement, make it short and discuss point by point. 

Response 3. The discussion section was almost completely modified following the reviewer's suggestions. It was divided point by point for better organization.

Point 4. Some minor points are in pdf. File

Response 4. All recommendations were attended.

Point 4.1. Next, the isolation of an individual colony was transferred to new PDA Petri dishes. How? Cut the hyphal tip and transferred to PDA?

Response 4.1. Yes, we cut the hyphal tip and transferred to PDA medium.

Point 4.2. Why the dilution of soil indicates as conidia/mL?

Response 4.2. This is a mistake. This part was corrected as follows: For isolates from the soil, 10 g were weighed and suspended in 90 mL of sterile distilled water. From this mixture, a 1:1000 v/v dilution was made.

Point 4.3. Twenty microliters of the dilution  were taken and placed on PDA with 0.025 mg/mL chloramphenicol.  Not clear.

Response 4.3. This part was corrected as follows: For isolates from the soil, 10 g were weighed and suspended in 90 mL of sterile distilled water. From this mixture, a 1:1000 v/v dilution was made. Twenty microliters of the dilution were taken and placed on PDA with 0.025 mg/mL chloramphenicol using the spread plate technique.

Point 4.4. Microscopic characteristics were traditionally examined. By which equipment? Which microsope? How about magnitude? And how many replicate n=?

Response 4.4. Microscopic characteristics (shape and size of conidiophores, phialides, conidia, and chlamydospores) were traditionally examined with an optical microscope (Model CX311RTSF, Olympus, Tokyo, Japan), using 40× magnification with 40 measurements. This was added to the document.

Point 4.5. Inhibition of radial mycelial growth (IRM) was calculated based on the following formula:  Need citation.

Response 4.5. The growth of both colonies (cm) was measured, and the percentage of inhibition of radial mycelial growth (IRM) was calculated based on the following formula according to Ibarra et al., 2010.

Ibarra-Medina VA, Ferrera-Cerrato R, Alarcon A, Lara-Hernández ME, Valdez-Carrasco JM. 2010. Isolation and screening of Trichoderma strains antagonistic to Sclerotia sclerotium and Sclerotia minor. Revista Mexicana de Micología. 31:53-63.

Point 4.6. Trichoderma spp. (1 x 106 conidia/mL) was inoculated in the peduncular area of ​​avocado fruit. Please specify volume used in each treatment.

Response 4.6. Trichoderma spp. (1 x 106 conidia/mL) was inoculated (25 mL) in the peduncular area of ​​avocado fruit.

Point 4.7. Chitinase and glucanase activities, why don't express as U/mL?

Response 4.7. The units of U/mL are more precise because they depend on the amount of protein (that is in the extract), however, for our study we are only interested in observing the effect (increase or decrease) of these enzymes on the lysis of the cell wall of pathogens, so most of the studies that were used to discuss our results are not reported in U/mL, for example in  Adeline Su YienTing and Jing Yun Chai. 2015. Chitinase and β-1,3-glucanase activities of Trichoderma harzianum in response towards pathogenic and non-pathogenic isolates: Early indications of compatibility in consortium. Biocatalysis and Agricultural Biotechnology. 4: 109-113

Reviewer 3 Report

Manuscript number: horticulturae-1821062

 Title: Isolation and characterization of Trichoderma spp. for antagonistic activity against avocado (Persea americana Mill) fruit pathogens

The manuscript presents interesting results that can contribute to the understanding of the postharvest application of Trichoderma harzianum in avocado. I believe the manuscript is of interest to Horticulturae readers. However, some important issues need to be clarified.

TITLE

The title is appropriate to the work performed.

ABSTRACT

The abstract is not adequate and needs an adjustment. Data from morphological and molecular characterization of the isolates must be placed after the isolation of the strains. The authors should add more information about the results obtained in the abstract.

KEYWORDS – Keywords is adequate and gives a good idea of the work done.

INTRODUCTION

In general, the introduction provides important information about the work with avocado. However, the authors do not describe the work already carried out with fungi of the species Trichoderma in post-harvest treatments of different fruits.

 MATERIALS AND METHODS

The methodology is well described and is sufficient to achieve the proposed objectives. Isolation and characterization of Trichoderma strains are adequate. Molecular markers were used, ITS regions and the elongation factor Tef1 α-1, as recommended in the literature.

Line 183: the figure 1 must be removed or added as a complementary data.

RESULTS

Line 239-315. The Isolation and identification of Trichoderma spp. strains were done properly, and the authors chose six strains classified as T. harzianum to continue the work. This choice was based on the antagonistic capacity of Trichoderma spp. The results showed the potential of these isolates as a biological control agent for the phytopathogens used.

The authors performed microscopy work to show the behavior of the hyphae of the T. harzianum strains and the phytopathogens used. These are interesting data showing that antagonist strains undergo thickening of hyphae, while the phytopathogen tends to thin, except for Diaporthe sp., which tends to thicken.

Line 359. It is unclear why the authors determined the activity of chitinase and glucanase enzymes only from the TSMICH7- strain. The idea was to relate the enzyme activity with the greater efficiency of this strain against phytopathogens? The authors tested whether the obtained cell extract had an effect on the cell wall lysis of phytopathogens?

Line 390. The data in figure six should be revised. I suggest authors only consider the 72-hour time, where activities are at their greatest. After this time of culture, lysis of cells can occur, and intracellular enzymes can be released into the culture medium.

DISCUSSION

This section is very long and repeats some information already presented in the results. The authors should compare the results obtained with data from the literature related to the use of Trichoderma in post-harvest treatments of various fruits. These fungi are recognized for their effectiveness in controlling postharvest disease in a range of fruits such as apple (Rhizopus stolonifera), banana (C. musae), citrus (Penicillium italicum), mango (C.  gleosporioides), as well as kiwifruit (Botrytis cinerea) and strawberry (Botrytis cinerea).

The authors must show the differential of the data obtained and the contribution they can make in the area.

Author Response

Thanks to the reviewer for all appropriate comments and suggestions. We attended all the observations according to each one of the comments.

Point 1. TITLE

Point 1.1.  The title is appropriate to the work performed.

Response 1.1. Thank you.

Point 2. ABSTRACT

Point 2.1.  The abstract is not adequate and needs an adjustment. Data from morphological and molecular characterization of the isolates must be placed after the isolation of the strains. The authors should add more information about the results obtained in the abstract.

Response 2.1. The abstract was modified according to the comments of the reviewer, considering that it should not exceed 200 words.

Point 3. KEYWORDS – Keywords is adequate and gives a good idea of the work done.

Response 3. Thank you.

Point 4. INTRODUCTION

Point 4.1.  In general, the introduction provides important information about the work with avocado. However, the authors do not describe the work already carried out with fungi of the species Trichoderma in post-harvest treatments of different fruits.

 Response 4.1. Different studies related to the use of Trichoderma in different fruits were added to the introduction.

Point 5. MATERIALS AND METHODS

Point 5.1. The methodology is well described and is sufficient to achieve the proposed objectives. Isolation and characterization of Trichoderma strains are adequate. Molecular markers were used, ITS regions and the elongation factor Tef1 α-1, as recommended in the literature.

Response 5.1. Thank you.

Point 5.2. Line 183: the figure 1 must be removed or added as a complementary data.

Response 5.2. This figure is available as supplementary material (Figure S1).

Point 6. RESULTS

Point 6.1. Line 239-315. The Isolation and identification of Trichoderma spp. strains were done properly, and the authors chose six strains classified as T. harzianum to continue the work. This choice was based on the antagonistic capacity of Trichoderma spp. The results showed the potential of these isolates as a biological control agent for the phytopathogens used.

Response 6.1. Thank you.

Point 6.2. The authors performed microscopy work to show the behavior of the hyphae of the T. harzianum strains and the phytopathogens used. These are interesting data showing that antagonist strains undergo thickening of hyphae, while the phytopathogen tends to thin, except for Diaporthe sp., which tends to thicken.

Response 6.2. Thank you.

Point 6.3. Line 359. It is unclear why the authors determined the activity of chitinase and glucanase enzymes only from the TSMICH7- strain. The idea was to relate the enzyme activity with the greater efficiency of this strain against phytopathogens?

Response 6.3. The best strain in the in vivo analysis was T. harzianum (TSMICH7). Hence, this strain was used to determine the reducing sugars and lytic activities (chitinase and glucanase) with each phytopathogens. This was indicated in the document.

Point 6.4. The authors tested whether the obtained cell extract had an effect on the cell wall lysis of phytopathogens?

 Respoonse 6.4. We use crude extracellular extracts to determine the effect on the cell wall lysis of phytopathogens through the measurement of glucananases and chitinases. If the amount of reducing sugars increased during the confrontation, it was considered an indication that the lytic enzymes (chitinases and glucanases) act on the pathogen's cell walls, degrading the chitin polymers and glucan. Glucanase and chitinase activities were determined as the difference between the production activity of reducing sugars without substrate minus the enzymatic activity of reducing sugars with the substrate. We carried out another study where the crude extract was added directly to the pathogens, however, the study was left unincluded due to pandemic situations and it was decided not to publish it on this occasion.

Point 6.5. Line 390. The data in figure six should be revised. I suggest authors only consider the 72-hour time, where activities are at their greatest. After this time of culture, lysis of cells can occur, and intracellular enzymes can be released into the culture medium.

Response 6.5. Figure 6 was modified for better organization and understanding. It was divided into four parts organizing it for each phytopathogen. We appreciate the suggestion of leaving it only up to 72 hours, however, for purposes of comparison with other studies and to explain their own results, most of the authors decided to leave it up to 168 hours.

Point 7. DISCUSSION

Point 7.1. This section is very long and repeats some information already presented in the results. The authors should compare the results obtained with data from the literature related to the use of Trichoderma in post-harvest treatments of various fruits. These fungi are recognized for their effectiveness in controlling postharvest disease in a range of fruits such as apple (Rhizopus stolonifera), banana (C. musae), citrus (Penicillium italicum), mango (C.  gleosporioides), as well as kiwifruit (Botrytis cinerea) and strawberry (Botrytis cinerea).

Response 7.1. The discussion section was almost completely modified following the reviewer's suggestions. It was divided point by point for better organization.

Point 7.2. The authors must show the differential of the data obtained and the contribution they can make in the area.

Response 7.2. The principal contribution of this research is the antagonistic capacity of Trichoderma  against P. perseae, N. parvum, C.  gleosporioides and Diaporthe sp. pathogens of avocado and therefore in its usefulness as a biopesticide or as a biological control to preserve the quality of this important international fruit. One of the interesting things about the study is that everything was isolated directly from avocado crops, so the specificity of the isolated species may be more powerful than using other isolates from other crops. By controlling these diseases, the quality and export of these fruits is more assured, consequently bringing better economy to both farmers and the producing country.

Reviewer 4 Report

Dear Authors,

This research is very interesting but the paper can't be accepted in the current form. In some parts it sounds very confusing and the english needs to be improve. I have some suggestion for you:

1) In the abstract you wrote that Trichoderma strains were isolated from leaf, roots, fruits and soil of avocado orchards. In Materials and Methods I only read about isolation from roots and soil. What about leaf and fruits?

2) All species names must be written in italic. Please pay attention.

3) "Macroscopic and microscopic characteristics were traditionally examined". How? Did you use a microscope for microscopic observations? I suppose yes, so you should write it.

4)Lines 168-171: this phrase is unclear.

5) Lines 213-215: any reference about this calculation?

6) Lines 300-310: why didn't you write something about this in M&M? It is a very interesting analisis.

7) Lines 360-361: you wrote that TSMICH7 was used to determine reducing sugars and lytic activities. So, why in Figure 6 you cited TRJAL7 strain?

8) Something went wrong with figures 5-6-7 and the results of enzymatic activity are confusing and hard to understand. The figure presented is written in spanish. Please correct it in english. 

In general I suggest to rewrite more clearly the results and revised all the manuscript paying attention to the english form.

Author Response

Thanks to the reviewer for all appropriate comments and suggestions. We attended all the observations according to each one of the comments.

Point 1. In the abstract you wrote that Trichoderma strains were isolated from leaf, roots, fruits and soil of avocado orchards. In Materials and Methods I only read about isolation from roots and soil. What about leaf and fruits?

Response 1.  Trichoderma strains were isolated from roots and soil. Leaf and fruits were a mistake. We corrected this.

Point 2. All species names must be written in italic. Please pay attention.

Response 2. All these errors were reviewed and corrected.

Point 3. "Macroscopic and microscopic characteristics were traditionally examined". How? Did you use a microscope for microscopic observations? I suppose yes, so you should write it.

Response 3. Macroscopic characteristics of the colony (radial growth, mycelial color, presence of diffusible pigments, concentric rings) and microscopic characteristics (shape and size of conidiophores, phialides, conidia, and chlamydospores) were traditionally examined with an optical microscope (Model CX311RTSF, Olympus, Tokyo, Japan), using 40× magni-fication with 40 measurements. This was added to the document.

Point 4. Lines 168-171: this phrase is unclear.

Response 4. This phrase was changed as follows:” TSMICH7 was the best strain that showed higher pathogen inhibitions in the in vivo test. For this reason, this Trichoderma strain was selected to obtain the extracellular extract and subsequently carry out the determinations of reducing sugars content and the lytic enzyme activities.”

Point 5. Lines 213-215: any reference about this calculation?

Response 5. In this part, Glucanase and chitinase activities were determined as the difference between the pro-duction activity of reducing sugars without substrate minus the enzymatic activity of reducing sugars with the substrate and from co-inoculations with their corresponding controls (single inoculations). The above is according to Adeline Su YienTing and Jing Yun Chai. 2015. Chitinase and β-1,3-glucanase activities of Trichoderma harzianum in response towards pathogenic and non-pathogenic isolates: Early indications of compatibility in consortium. Biocatalysis and Agricultural Biotechnology. 4: 109-113.

Point 6. Lines 300-310: why didn't you write something about this in M&M? It is a very interesting analisis.

Response 6. We described the morphometric study in materials and methods. “Additionally, we conducted a morphometric study to understand the microscopical changes in the structure of both organisms when confronting them. This study consisted of determining the average diameter of hyphae before and after confrontation (Tricho-derma/Phytopathogen) using an optical microscope (Model CX311RTSF, Olympus, Tokyo, Japan) integrated into an Infinity 1 camera (Lumenera Corp., Ottawa, ON, Canada). This assay was taken with a 40× objective and analyzed using Image Pro-Plus ver. 6.3 software (Media Cybernetics, Inc., Bethesda, MD, USA). At least 60–100 measurements of the diameter of the hyphae were measured from each treatment.”

Point 7.  Lines 360-361: you wrote that TSMICH7 was used to determine reducing sugars and lytic activities. So, why in Figure 6 you cited TRJAL7 strain?

Response 7. This was a mistake. The correct sample is TSMICH7. This was corrected and figure 6 was modified.

Point 8. Something went wrong with figures 5-6-7 and the results of enzymatic activity are confusing and hard to understand. The figure presented is written in spanish. Please correct it in english. 

Response 8. We organized better the results, mainly everything related to the figures 5-6-7 which were divided each one in four parts indicating A: assay with N. parvum; B: assay with C. gloeosporioides; C: assay with Diaporthe sp., and D: assay with P. perseae. All figures are in English.

Point 9. In general I suggest to rewrite more clearly the results and revised all the manuscript paying attention to the english form.

Response 9. We tried to organize better the results.  The English form was sent to be reviewed by experts from the MDPI publisher.

Point 10. Comments in the pdf. File

Response 10. All recommendations were attended.

Point 10.1. Isolation of Trichoderma from leaf and fruit is not present in Materials and Methods.

Point 10.1. Trichoderma strains were isolated from roots and soil. Leaf and fruits were a mistake. We corrected this.

Point 10.2. Next, the isolation of an individual colony was transferred to new PDA Petri dishes. This is unclear. How? Monosporic culture? Pure colonies from hyphal tips? Specify.

Response 10.2. From the growths obtained, monosporic cultures were made from hyphal tips, obtaining from this process an axenic strain established in a new Petri dish containing PDA.

Point 10.3. Finally, serial dilutions were made until 1 x 103 conidia/mL was obtained. Why the dilution of soil indicates as conidia/mL?

Point 10.3. This is a mistake. This part was corrected as follows: For isolates from the soil, 10 g were weighed and suspended in 90 mL of sterile distilled water. From this mixture, a 1:1000 v/v dilution was made. Twenty microliters of the dilution were taken and placed on PDA with 0.025 mg/mL chloramphenicol using the spread plate technique.

Point 10.4. How do you place soil suspension in PDA? Did you use the spread plate technique? Specify.

Response 10.4. Twenty microliters of the dilution were taken and placed on PDA with 0.025 mg/mL chloramphenicol using the spread plate technique.

Point 10.5. What do you mean by traditionally examined? Did you use a microscope? Indicate which microscope. Which magnitude? How many measurements did you take?

Response 10.5. Macroscopic characteristics of the colony (radial growth, mycelial color, presence of diffusible pigments, concentric rings) and microscopic characteristics (shape and size of conidiophores, phialides, conidia, and chlamydospores) were traditionally examined with an optical microscope (Model CX311RTSF, Olympus, Tokyo, Japan), using 40× magni-fication with 40 measurements.

Point 10.6. What do you mean by recovered? Maybe observed?

Response 10.6. The amplified DNA fragments were separated and observed in 2% agarose gel.

Point 10.7. Did you perform a phylogenetic analysis?

Response 10.7. Yes we did a phylogenetic analysis, it is as figure 1.

Point 10.8. Why do you use this formula? There are any references?

Response 10.8. We are based in the reference Ibarra-Medina VA, Ferrera-Cerrato R, Alarcon A, Lara-Hernández ME, Valdez-Carrasco JM. 2010. Isolation and screening of Trichoderma strains antagonistic to Sclerotia sclerotium and Sclerotia minor. Revista Mexicana de Micología. 31:53-63.

Point 10.9. Why didn't you write about morphometric analysis in the M&M?

Response 10.9. We described the morphometric study in materials and methods. “Additionally, we conducted a morphometric study to understand the microscopical changes in the structure of both organisms when confronting them. This study consisted of determining the average diameter of hyphae before and after confrontation (Tricho-derma/Phytopathogen) using an optical microscope (Model CX311RTSF, Olympus, Tokyo, Japan) integrated into an Infinity 1 camera (Lumenera Corp., Ottawa, ON, Canada). This assay was taken with a 40× objective and analyzed using Image Pro-Plus ver. 6.3 software (Media Cybernetics, Inc., Bethesda, MD, USA). At least 60–100 measurements of the diameter of the hyphae were measured from each treatment.”

Point 10.10. If only Trichoderma harzianum TSMICH7 was used to determine enzymatic activity as you report in M&M, why now you write here the activity of the strain TRJAL7?

Response 10.10. This was a mistake. The correct sample is TSMICH7. This was corrected and figure 6 was modified.

Round 2

Reviewer 2 Report

The authors carefully revised this manuscript according to reviewer comments.

Reviewer 3 Report

Manuscript number: horticulturae-1821062

 Title: Isolation and characterization of Trichoderma spp. for antagonistic activity against avocado (Persea americana Mill) fruit pathogens

 The manuscript presents interesting results that may contribute to the understanding of the interaction between Trichoderma harzianum and avocado (Persea americana Mill) fruit pathogens. My suggestions in the first review were accepted by the authors and the manuscript is now better presented. Therefore, the manuscript is ready to be published. I believe the manuscript is of interest to Horticulturea readers.

Reviewer 4 Report

Dear Authors,

thank you for making the required changes. The article can now be accepted.